# Effect of Microstructure on the Onset Strain and Rate per Strain of Deformation-Induced Martensite Transformation in Q&P Steel by Modeling

**DOI:** 10.3390/ma15030952

**Published:** 2022-01-26

**Authors:** Jingyi Cao, Jianfeng Jin, Shaojie Li, Mingtao Wang, Shuai Tang, Qing Peng, Yaping Zong

**Affiliations:** 1School of Materials Science and Engineering, Northeastern University, Shenyang 110819, China; neu_caojy@163.com (J.C.); lisj_neu@163.com (S.L.); wangmingtao@mail.neu.edu.cn (M.W.); ypzong@mail.neu.edu.cn (Y.Z.); 2State Key Laboratory of Rolling and Automation, Northeastern University, Shenyang 110819, China; tangshuai@ral.neu.edu.cn; 3State Key Laboratory of Nonlinear Mechanics, Institute of Mechanics, Chinese Academy of Sciences, Beijing 100190, China; pengqing@imech.ac.cn

**Keywords:** deformation-induced martensitic transformation, microstructure, onset strain of phase transformation, Q&P steel, micromechanics model

## Abstract

The effect of microstructure on the onset strain and rate of deformation-induced martensitic transformation (DIMT) in Q&P steel is studied by a mean-field micromechanics model, in which the residual austenite (*RA*) and primary martensite (*M*) phases are treated as elastoplastic particles embedded into the ferrite (*F*) matrix. The results show that when the volume fraction of the *RA* increases with a constant fraction of the *M*, the onset strain of DIMT increases and transformation rate decreases, in contrast to the case of the *RA* fraction effect with a fixed *F* fraction. Increasing the volume fraction of the *M* postpones the DIMT, regardless of the corresponding change from the *RA* or *F* fraction, which is similar to the effect of the *RA* fraction with the constant *M* but to a higher degree. Conversely, when increasing the fraction of the matrix *F*, the onset strain of DIMT increases and the rate decreases, and the effect is greater when the corresponding fraction change comes from the *M* rather than from the *RA*. Moreover, when the aspect ratio of the *RA* increases, the onset strain of DIMT decreases with a gradual increase in transformation rate, in agreement with the experimental observation that the equiaxial austenite is more stable in Q&P steels. However, the aspect ratio effect of the *M* is opposite to that of the *RA*, indicating that the lath-shaped primary martensite could protect the austenite from DIMT.

## 1. Introduction

As one of the typical advanced high-strength steels (AHSSs), quenching and partitioning (Q&P) steel can be used in vehicle light-weighting, energy saving, and emission reduction due to its excellent strength and ductility [1]. After quenching and partitioning processing, Q&P steel has a complex microstructure composed of ferrite, residual austenite, and martensite [2]. Mechanical behaviors of Q&P steel are significantly affected by the residual austenite phase due to deformation-induced martensite transformation (DIMT). As the steel for automobile components, the occurrence of DIMT in Q&P steel can greatly improve driving safety by absorbing more kinetic energy during impacting [3]. Therefore, understanding the effect of microstructure on the DIMT in Q&P steel is essential to control the occurrence of DIMT in service.

In multiphase steels, the residual austenite is observed in different morphologies, such as blocky, film-like, or inter-lath lamellar shapes [4,5,6]. Each morphology has its own characteristics in the DIMT. During deformation, the blocky austenite grain usually has a low transformation stability [4], the lamellar-shaped one has intermediate stability, and the film-like austenite shows the highest stability against the DIMT [5]. Moreover, it is reported that the lamellar austenite grain with a larger aspect ratio is less stable than one with a smaller ratio [6]. In addition, plastic deformation behaviors of the surrounding phases of the austenite have also been reported to affect its transformation stability [7,8]. Therefore, the stability of residual austenite depends not only on its own characteristics, but also on the microstructural features such as the shape and content of other phases in Q&P steel. Furthermore, during hot forming of martensitic stainless steels, this microstructure effect on the features of DIMT is also important to optimize processing parameters and mechanical properties of the steels [9].

In order to understand and manipulate the DIMT in multiphase steels, the effects of microstructural factors on the DIMT need to be clarified, in which the accurate stress and strain partitioning in the residual phase is required. In some works [10,11,12,13,14], the effects of the DIMT features on deformation behaviors of multiphase steels can be studied by modeling. For instance, the macroscopic behaviors of multiphase TRIP steel were predicted and the effects of phase composition and morphology on flow stress and strain hardening were studied from a physical-based model [15], in which the results showed that the much harder martensite phase gives rise to a strong hardening of the residual austenite islands. In the works [11,16,17], the stress and strain partitioning between different phases in multiphase TRIP steel was evaluated using a mean-field homogenization model and the results indicated that the optimum strength–ductility balance could be achieved by an intermediate stable austenite, where too rapid transformation of austenite led to a lack of hardening or too slow transformation could not provide the extra strain hardening required to postpone localization. Delannay et al. calculated the partitioning of strain between softer and harder constituents in a TRIP multiphase steel based on an elastoplastic Mori–Tanaka model, and the macroscopic stress–strain curves and rate of martensite formation subjected to various loading were predicted [12], indicating that the accommodation of the transformation strain had a negligible effect on macroscopic stress when the fraction of residual austenite was low. Moreover, two- and three-dimensional microstructure-based representative volume element models are also used to study the mechanical response of multiphase steels [18,19,20,21]. Choi et al. developed a microstructure-based finite element (FE) model and studied the failure mode of multiphase TRIP steel containing residual austenite with DIMT [18]. In [19], the effects of phase properties on strength and ductility in Q&P steel were examined by an FE model and it was found that when the stability and/or volume fraction of residual austenite were higher and the phase strength difference was less, the higher phase-hardening benefited Q&P steels. Srivastava et al. studied the effect of stress partitioning among phases on mechanical behaviors in Q&P980 steel [22], using a microstructure-based finite element model. However, few works focus on the onset of DIMT, which is critical for controlling DIMT in service.

In this work, using a mean-field micromechanics model [23,24], regarding residual austenite and primary martensite phases as the elastoplastic particulate reinforcements embedded into the ferrite matrix, the microstructural effect on onset strain and rate of deformation-induced martensitic transformation (DIMT) in Q&P steel is quantitively studied and compared, aiming to facilitate the development of AHSSs.

## 2. Micromechanics Model for Multiphase Q&P Steel

### 2.1. Modeling of Multiphase Q&P Steel

In this work, the material sample is commercial Q&P980 sheet steel as received from Baosteel in China with ultimate tensile strength of 980 MPa and thickness of 1.2 mm [19,25]. The chemical composition is listed in Table 1 and the microstructure contains ferrite, primary martensite, and residual austenite phases with volume fractions of 37%, 52%, and 11%, respectively, as shown in Figure 1a.

Based on an Eshelby model, a micromechanics model was established in our previous work to simulate the onset strain of DIMT in Q&P980 steel [24]. The model setup is schematically shown in Figure 1b. The assumptions in the micromechanics model are as follows: (1) the ferrite (*F*) was taken as the matrix, the residual austenite (*RA*) and martensite (*M*) were taken as the 1st and 2nd inhomogeneities, embedded in the ferrite matrix, (2) the averaged shapes of martensite and residual austenite are treated as prolate spheroids with aspect ratios $s\;\left(=a_{3}/a_{1}\right)$ of 3 and 1.2, respectively, and (3) the matrix–inhomogeneity interfaces are perfectly bonded.

In this work, the model was originally based on Eshelby’s equivalent inclusion approach [16] with the unique equation expressed as $C_{I}\left(e^{C}-e^{T*}\right)=C_{m}\left(e^{C}-e^{T}\right)$ [23], where $e^{C}$,$\;e^{T*}$, and $e^{T}$ are the constrained strain, mismatch strain, and imagined strain, respectively, $e^{C}=Se^{T}$ (S is Eshelby tensor), and $C_{I}$, and $C_{m}$ are the stiffness of inclusion (i.e., particle) and matrix, respectively. The equation built a relationship between the matrix and inclusions into a continuum mechanics framework. After combining the Mori–Tanaka mean-field method [17], a micromechanics model was developed and the stress in inclusions in a finite matrix was calculated by $\sigma_{I}=\sigma_{m}+C_{m}\left(e^{0}+e_{I}^{C}-e_{I}^{T}\right)+C_{m}\left(\bar{e}+e_{I}^{C}-e_{I}^{T}\right)$ [23], where $\sigma_{m}$ is the matrix stress, $e^{0}$ is a far-field strain of the matrix, and $\bar{e}$ is the interactive strain in the matrix. The advantage of the Eshelby-based mean-field model is that the perturbation effect of inhomogeneities embedded in the matrix is considered and background stress in the matrix is introduced to account for the disturbance of the matrix elastic strain fields surrounding the inhomogeneities neighboring each other in a finite, non-diluted system, namely, the mean strain within the matrix from the elastic stress field of homogeneous and heterogeneous materials is discrepant. More details about the model were described in previous works [23,24] and this work focuses on the analysis of the simulation results and potential applications in the steel industry.

### 2.2. Constitutive Inputs of Each Phase

The elastic modulus of ferrite, residual austenite, and primary martensite are 220 GPa, 187 GPa, and 187 GPa, respectively, and Poisson’s ratio of each phase is 0.3 [12,13]. The experimental stress–strain curves [12,18,20] were used as the constitutive inputs to describe elastoplastic responses of the individual phases of residual austenite and primary martensite, which were validated in our previous work [24]. After comparing the experimental stress–strain curves of the single-phase ferrite steels in a tensile test [18,19,20,21] in Figure 2a, the constitutive input [19] is chosen as the input of the matrix ferrite phase, due to a similar chemical composition of the experimental Q&P980 sample used in this work.

The macroscopic stress–strain curve predicted by the micromechanics model is shown in Figure 2b, and is in good agreement with the experimental one of Q&P980 steel during a tensile test [25]. The difference between the predicted and input stress of the residual austenite represents the load bearing in multiphase Q&P980 steel, which indicates that the model can be attainable to study the DIMT characteristics based on the austenite stress partitioning above.

## 3. Results and Discussion

### 3.1. Prediction of Onset Strain and Tranformation Rate of DIMT in Q&P980 Steel under Uniaxial Loading

In our previous work [24], it was indicated that the onset strain of DIMT can be determined by a stress criterion of martensite transformation as a function of the applied strain. The calculated critical stress of the onset of the martensite transformation was about 985 MPa [24] and the onset strain of DIMT ($\varepsilon_{onset}$) was the corresponding applied strain in the predicted stress–strain curve of residual austenite when the stress ($\sigma_{RA}$) was equal to 985 MPa. Since the stress partitioning of residual austenite varies with microstructure, the effects of the microstructure of the Q&P steel on onset strain of DIMT can be examined by the micromechanics model under a far-field applied strain.

In order to calculate the transformation rate of the DIMT, based on the experimental results of Q&P steels [25,26,27], the transformation proportion of DIMT $\left(f_{\mathrm{DIMT}}=f_{NB}/f_{RA}^{}\right)$ is fitted out as a function of applied strain in Figure 3,
(1)$$f_{\mathrm{DIMT}}\left(\varepsilon \right)=27.6\;e^{2.4\varepsilon }-45.2\;e^{-30.4\varepsilon }$$
where $f_{NB}$ and $f_{RA}^{}$ are the volume fraction of newborn martensite from DIMT and original residual austenite, respectively. Here, it is assumed that there is no microstructural effect on the stress criterion of martensite transformation.

Combining the predicted stress–strain curve of the *RA* in Figure 2b with Equation (1), we can deduce a relationship between $f_{\mathrm{DIMT}}$ and the stress in residual austenite ($\sigma_{RA}$) as
(2)$$f_{\mathrm{DIMT}}\left(\sigma_{RA}\left(\varepsilon \right)\right)=1.33\times {\left(\sigma_{RA}-985\right)}^{0.54}$$

In this work, the rate per strain of DIMT is presented by the $f_{\mathrm{DIMT}}$ at the applied strain of 15%.

### 3.2. Volume Fraction Effect of Each Constitutive Phase on Onset Strain and Rate of DIMT during Tensile Loading

In this section, the volume fraction effect of each constitutive phase on the onset strain ($\varepsilon_{\mathrm{onset}}$) and rate of DIMT ($f_{\mathrm{DIMT}}$ at an applied strain of 15%) during tensile loading is predicted, where the change in volume fraction is in the range of 2~41% for residual austenite (*RA*), 32~62% for primary martensite (*M*), and 17~47% for ferrite phase (*F*).

Figure 4 illustrates the effect of the volume fraction of residual austenite (*f_RA_*) on onset strain and rate of DIMT, where the volume fractions of martensite (*f_M_*) and ferrite (*f_F_*) are fixed separately. When the *f_RA_* increases from 2% to 41% with *f_M_* of 52%, the $\varepsilon_{\mathrm{onset}}$ increases from 2.1% to 2.5% and $f_{\mathrm{DIMT}}$ decreases from 39.8% to 38.2%, as shown in Figure 4a, while in the case of the RA fraction effect with constant *f_F_* of 37% in Figure 4b, there is a sharp drop in the $\varepsilon_{\mathrm{onset}}$ and a steep rise in $f_{\mathrm{DIMT}}$, where the $\varepsilon_{\mathrm{onset}}$ falls to 0.8% from 2.8% and the $f_{\mathrm{DIMT}}$ rises to a high point at 46.1% from 37.3% with *f_RA_* from 2% to 41%. The results suggest that as the *RA* proportion increases, mechanical stability of the *RA* becomes higher when the ferrite matrix decreases, while it becomes less stable when the harder primary martensite decreases.

Figure 5 shows the effect of the volume fraction of martensite (*f_M_*) on onset strain and rate of DIMT, corresponding to the change in *f_RA_* or *f_F_*, respectively. When the *f_M_* increases from 32% to 62% with a constant *f_RA_* of 11%, the $\varepsilon_{\mathrm{onset}}$ increases from 1.0% to 2.9% and $f_{\mathrm{DIMT}}$ decreases from 44.8% to 36.9%, as shown in Figure 5a. When the *f_M_* increases from 32% to 62% with a constant *f_F_* of 37% in Figure 5b, the $\varepsilon_{\mathrm{onset}}$ rises to 2.8% from 1.2% and the $f_{\mathrm{DIMT}}$ falls to a low point at 37.3% from 43.8%. The results suggest that as the *M* proportion increases, mechanical stability of the *RA* becomes higher regardless the corresponding change from the *RA* or *F* fraction.

Figure 6 presents the effect of the *f_F_* on onset strain and rate of DIMT, where the *f_M_* and *f_RA_* are fixed separately. When the *f_F_* increases from 17% to 47% with a constant *f_M_* of 52%, the $\varepsilon_{\mathrm{onset}}$ decreases from 2.4% to 2.1% and $f_{\mathrm{DIMT}}$ increases from 38.6% to 39.8%, as in Figure 6a. In Figure 6b, the $\varepsilon_{\mathrm{onset}}$ falls to 1.6% from 3.7% and the $f_{\mathrm{DIMT}}$ rises to a high value at 42.1% from 34.5%, when the *f_F_* is from 17% to 47% with a constant *f_RA_* of 11%. The results suggest that as the *F* proportion increases, mechanical stability of the *RA* becomes lower, associated with the decrease in *RA*. A similar but more significant effect of the *F* fraction is observed when the corresponding *M* fraction decreases.

### 3.3. Aspect Ratio Effect of Particle-Shaped Residual Austenite and Primary Martensite on Onset Strain and Rate of DIMT during Tensile Loading

In this section, the onset strain and rate of DIMT are calculated and compared with different aspect ratios of the residual austenite (*RA*) and primary martensite (*M*) of Q&P steel, respectively.

In Figure 7, the effect of the *RA* aspect ratio along the tensile direction on the $\varepsilon_{\mathrm{onset}}$ and $f_{\mathrm{DIMT}}$ is studied, with a constant *M* aspect ratio of 3.0. The $\varepsilon_{\mathrm{onset}}$ decreases from 3.5% to 2.4% and the $f_{\mathrm{DIMT}}$ decreases from 34.5% to 38.7%, when the *RA* aspect ratio changes from 1/9 to 1 in Figure 7a. The $\varepsilon_{\mathrm{onset}}$ further decreases from 2.4% to 1.1% and the $f_{\mathrm{DIMT}}$ increases from 38.7% to 45.1% when the RA aspect ratio varies from 1 to 9 in Figure 7b. The results indicate when the aspect ratio of the *RA* increases, the onset strain of DIMT decreases with a gradual increase in transformation rate, in agreement with the experimental observation that the equiaxial austenite is more stable in Q&P steels [6].

Figure 8 shows the effect of the *M* aspect ratio on the $\varepsilon_{\mathrm{onset}}$ and $f_{\mathrm{DIMT}}$, with the *RA* aspect ratio being 1.2. In Figure 8a, the $\varepsilon_{\mathrm{onset}}$ increases from 0.3% to 0.7% and $f_{\mathrm{DIMT}}$ decreases from 53.1% to 47.2% when the *M* aspect ratio changes from 1/9 to 1. Figure 8b shows that the $\varepsilon_{\mathrm{onset}}$ increases from 0.7% to 4.1% and the $f_{\mathrm{DIMT}}$ decreases from 47.2% to 34.1% when the *M* aspect ratio further changes from 1 to 9. The results indicate that the aspect ratio effect of the *M* is opposite to that of the *RA* and the lath-shaped primary martensite could protect the residual austenite from DIMT.

### 3.4. Applications of the Modeling Results in Controlling DIMT of Q&P Steel

The results above give better understanding on the effects of microstructural factors on DIMT in Q&P steel, which can provide some guidance to design the chemical composition, optimize processing parameters, and improve in-service performance. For instance, since the increase in the *RA* fraction may be attributed to an increase in the carbon content in Q&P steel [28], the effect of carbon content on DIMT can be related to the findings in Figure 4. According to the experimental observations in the process of Q&P steel [29,30], increasing quenching temperature may lead to a decrease in the martensite proportion, in which the volume fraction of martensite in a medium-carbon Q&P steel decreased from 80% to 26% with quenching temperature (QT) from 200 to 320 °C [29], while the martensite fraction in a low-carbon Q&P steel decreased from 82% to 36% within the same range of QT [30]. In additional, the average carbon content of the residual austenite appeared to be almost independent of QT. Therefore, the results in Figure 5 could facilitate the understanding of the effect of quenching temperature on DIMT in Q&P steel. Moreover, it is also suggested that inputting different constitutive relationships of the phase into the micromechanics model may represent the characteristics of the phase, such as alloying [28] or loading conditions [31] in Q&P steel, so more benefits can be achieved through further modeling efforts.

## 4. Conclusions

Microstructure effect on onset strain and transformation rate of deformation-induced martensitic transformation (DIMT) in Q&P steel can be quantitatively studied through a micromechanics model.When the volume fraction of residual austenite (*RA*) increases from 2% to 41% with the martensite fraction of 52%, the onset strain of DIMT increases from 2.1% to 2.5% and transformation rate decreases from 39.8% to 38.2%, in contrast to the case of the *RA* fraction effect with the constant fraction of ferrite.With the volume fraction of the primary martensite (*M*) increasing from 32% to 62%, the DIMT can be postponed, regardless of the corresponding change from the *RA* or *F* fraction, where the onset strain of DIMT increases from 1.0% to 2.9% (with a constant fraction of *RA* (*f_RA_* = 11%) or from 1.2% to 2.8% (*f_F_* = 37%), and the transformation rate decreases from 44.8% to 36.9% (*f_RA_* = 11%) or from 43.8% to 37.3% (*f_F_* = 37%).When increasing the volume fraction of the matrix ferrite from 17% to 47%, the onset strain of DIMT decreases from 2.4% to 2.1% (*f_M_* = 52%) or from 3.7% to 1.6% (*f_RA_* = 11%) and the transformation rate increases from 38.6% to 39.8% (*f_M_* = 52%) or from 34.5% to 42.1% (*f_RA_* = 11%).When the aspect ratio of the residual austenite increases from 1/9 to 9, the onset strain of DIMT decreases from 3.5% to 1.1% and transformation rate increases from 34.5% to 45.1%.When the aspect ratio of the primary martensite increases from 1/9 to 9, the onset strain of DIMT increases from 0.3% to 4.1% and transformation rate decreases from 53.1% to 34.1%.

## Figures and Tables

**Figure 1 materials-15-00952-f001:**
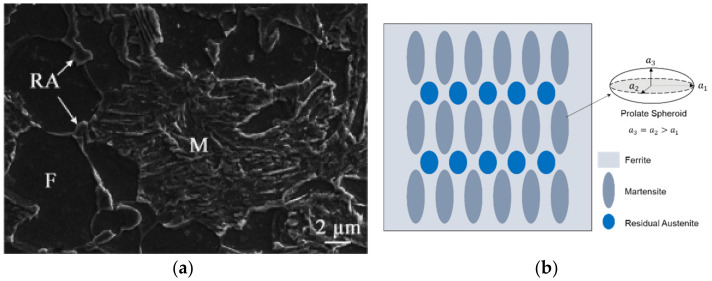
Microstructure of Q&P steel (**a**) from scanning electron microscopy [25] and (**b**) schematics of the mean-field micromechanics model setup.

**Figure 2 materials-15-00952-f002:**
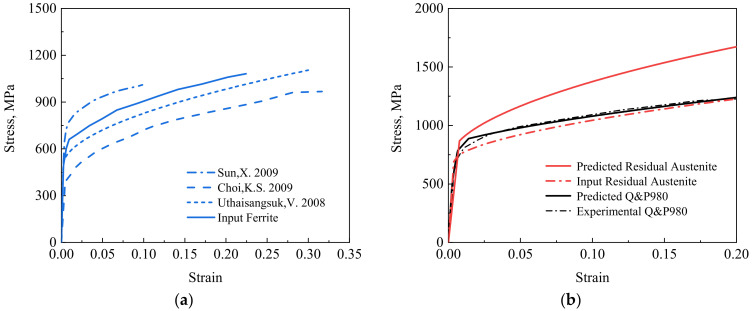
(**a**) Comparison of the experimental stress–strain curves of the single-phase ferrite steels [18,19,20,21] for choosing the input stress of the ferrite phase in the model of Q&P980 steel and (**b**) the predicted stress of Q&P980 steel and the stress partitioning in the residual austenite phase, in comparison to the experimental results of Q&P980 steel during a tensile test [25] and the constitutive input stress of residual austenite phase [24].

**Figure 3 materials-15-00952-f003:**
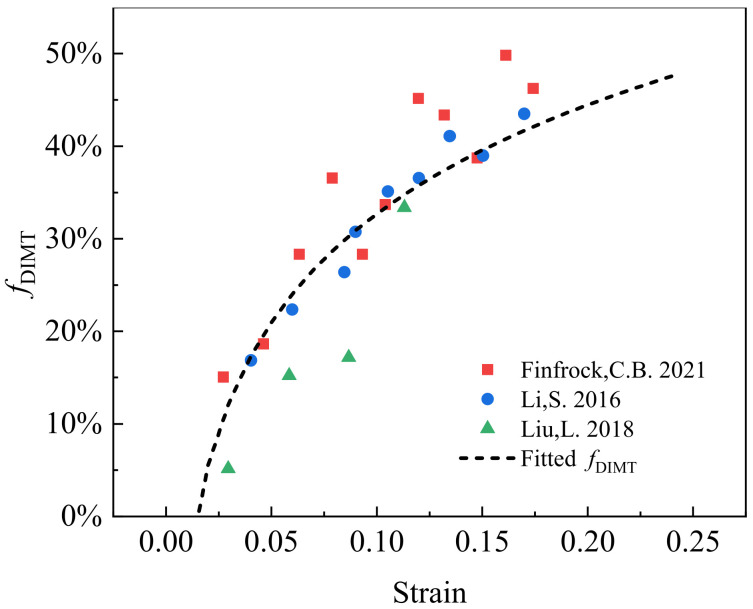
The transformation proportion of deformation-induced martensite transformation (DIMT) $\left(f_{\mathrm{DIMT}}=f_{NB}/f_{RA}^{}\right)$ during straining from the experiments in Q&P steels [25,26,27], where $f_{NB}$ and $f_{RA}^{}$ are volume fraction of the newborn martensite and initial residual austenite, respectively.

**Figure 4 materials-15-00952-f004:**
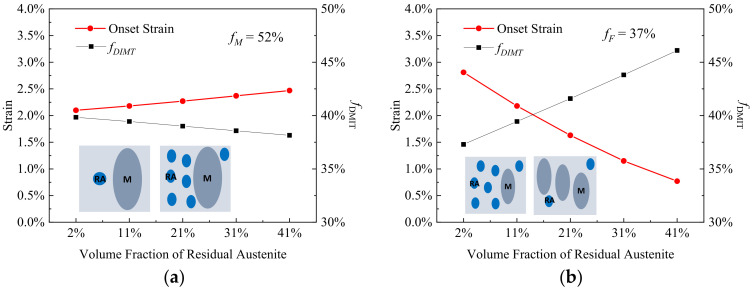
The onset strain and transformation rate of DIMT (*f_DIMT_* at the applied strain of 15%) during straining as a function of volume fraction of residual austenite (**a**) with the fixed volume fraction of primary martensite (*f_M_*) of 52% and (**b**) with the fixed volume fraction of ferrite (*f_F_*) of 37%, where the blue is remarked as the residual austenite and the dark gray is primary martensite within the insets.

**Figure 5 materials-15-00952-f005:**
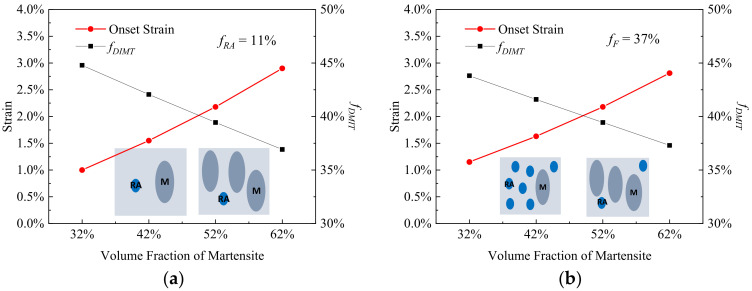
The onset strain and transformation rate of DIMT (*f_DIMT_* at the applied strain of 15%) during straining as a function of volume fraction of primary martensite (**a**) with the fixed volume fraction of residual austenite (*f_RA_*) of 11% and (**b**) with the fixed volume fraction of ferrite (*f_F_*) of 37%, where the blue is remarked as the residual austenite and the dark gray is primary martensite within the insets.

**Figure 6 materials-15-00952-f006:**
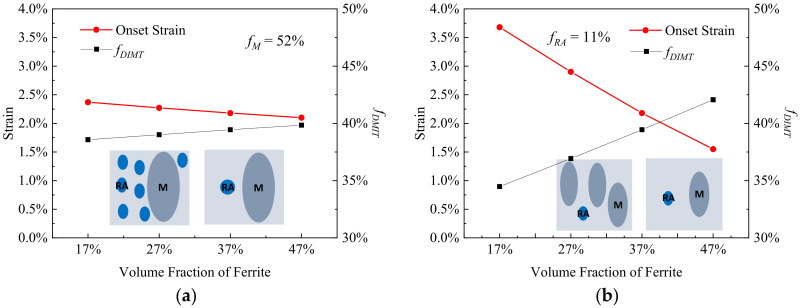
The onset strain and transformation rate of DIMT (*f_DIMT_* at the applied strain of 15%) during straining as a function of volume fraction of ferrite (**a**) with the fixed volume fraction of primary martensite (*f_M_*) of 52% and (**b**) with the fixed volume fraction of residual austenite (*f_RA_*) of 11%, where the blue is remarked as the residual austenite and the dark gray is primary martensite within the insets.

**Figure 7 materials-15-00952-f007:**
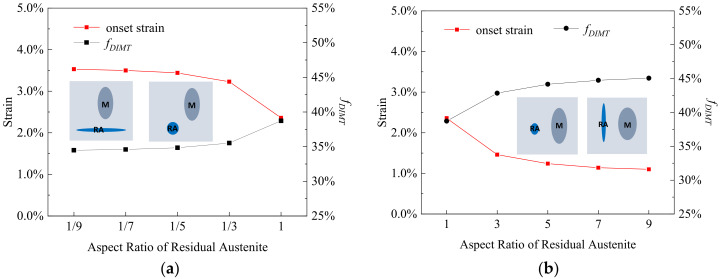
The onset strain and transformation rate of DIMT (*f_DIMT_* at the applied strain of 15%) during straining as a function of aspect ratio of residual austenite with the aspect ratio (**a**) from 1/9 to 1 and (**b**) from 1 to 9, where the blue is remarked as the residual austenite and the dark gray is primary martensite within the insets.

**Figure 8 materials-15-00952-f008:**
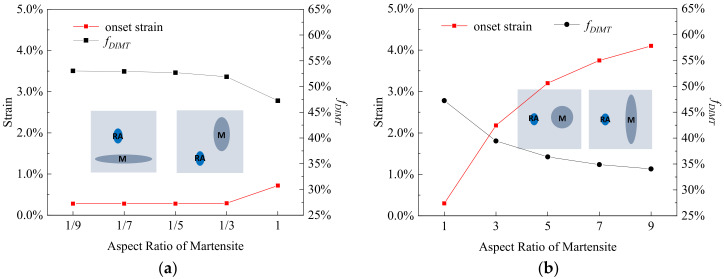
The onset strain and transformation rate of DIMT (*f_DIMT_* at the applied strain of 15%) during straining as a function of the aspect ratio of primary martensite with the aspect ratio (**a**) from 1/9 to 1 and (**b**) from 1 to 9, where the blue is remarked as the residual austenite and the dark gray is primary martensite within the insets.

**Table 1 materials-15-00952-t001:** Chemical composition of the samples of Q&P980 steel (wt%).

Element Type	C	Mn	Si	P	S	Al
Content	0.2	1.93	1.59	0.019	0.003	0.055

## Data Availability

Not applicable.

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
