# Peer review of "Effect of Microstructure on the Onset Strain and Rate per Strain of Deformation-Induced Martensite Transformation in Q&P Steel by Modeling"

_materials, 2022, doi:10.3390/ma15030952_

Round 1

Reviewer 1 Report

In this paper authors performed significant amount of the original scientific research on Q&P steel type.

In Introduction authors presented general state of research on this field and gave enough data for development of valid study and research.

However, a several issues is proposed to be discussed more precisely:

  • In Line “assumed that there is no microstructural effect on the stress criterion of martensite transformation“ authors assumed that there will be no microstructural effect. Is there a model that can incorporate influence of this effect on stress?
  • In Line „the matrix-inhomogeneity interfaces are perfectly bonded“, authors rely on the fact that there is perfect bond between matrix and inhomogeneities. Is there some experimental investigation that could give better understanding of the model and real material behaviour  which will incorporate actual quantity of inhomogeneities?
  • When speaking about application of this research; is there possible to give more detailed explanation of line “Therefore, the results in Figure 5 could facilitate the understanding about the effect of quenching temperature on DIMT in Q&P steel“ that is to say, how temperature is incoroporated in presented results?

Reviewer 2 Report

The manuscript deal with “Effect of microstructure on onset strain and rate per strain of deformation-induced martensite transformation in Q&P steel by modeling”. The topic is interesting and good results presented as well. There are a couple of comments that authors should revise before final decision:

1- The introduction should focused on modelling of microstructure of martensitic steel. Also the authors presented literature in this regard, but still a couple of literature available that authors should addressed in this section. Like: https://doi.org/10.3390/met11040572.

2- The referencing should be according to journal format.

3-   Please use complete word before abbreviation. For example 2-D/3-D in introduction.

4- Please present more details about used material. Also, it is better use a table to present chemical composition of Q&P steel.

5- Modelling method (what kind of modeling approach used in this article?) and used equations missed in the manuscript.

Round 2

Reviewer 2 Report

The authors are addressed the comments on good way